# Blood–Brain Barrier, Lymphatic Clearance, and Recovery: Ariadne’s Thread in Labyrinths of Hypotheses

**DOI:** 10.3390/ijms19123818

**Published:** 2018-11-30

**Authors:** Oxana Semyachkina-Glushkovskaya, Dmitry Postnov, Jürgen Kurths

**Affiliations:** 1Department of Human and Animal Physiology, Saratov State University, Astrakhanskaya Str. 83, 410012 Saratov, Russia; juergen.kurths@pik-potsdam.de; 2Department of Optics and Biophotonics, Saratov State University, 83 Astrakhanskaya str., 410012 Saratov, Russia; postnov@info.sgu.ru; 3Physics Department, Humboldt University, Newtonstrasse 15, 12489 Berlin, Germany; 4Potsdam Institute for Climate Impact Research, Telegrafenberg A31, 14473 Potsdam, Germany

**Keywords:** peripheral and meningeal lymphatics, blood-brain barrier, neurorehabilitation

## Abstract

The peripheral lymphatic system plays a crucial role in the recovery mechanisms after many pathological changes, such as infection, trauma, vascular, or metabolic diseases. The lymphatic clearance of different tissues from waste products, viruses, bacteria, and toxic proteins significantly contributes to the correspondent recovery processes. However, understanding of the cerebral lymphatic functions is a challenging problem. The exploration of mechanisms of lymphatic communication with brain fluids as well as the role of the lymphatic system in brain drainage, clearance, and recovery is still in its infancy. Here we review novel concepts on the anatomy and physiology of the lymphatics in the brain, which warrant a substantial revision of our knowledge about the role of lymphatics in the rehabilitation of the brain functions after neural pathologies. We discuss a new vision on the connective bridge between the opening of a blood–brain barrier and activation of the meningeal lymphatic clearance. The ability to stimulate the lymph flow in the brain, is likely to play an important role in developing future innovative strategies in neurorehabilitation therapy.

## 1. Lymphatics and Central Nervous System Health

Over the last few decades, the lymphatic mechanism underlying brain clearance and recovery has been actively debated [1,2,3,4,5,6,7,8,9,10,11,12,13,14,15,16,17,18,19,20,21,22,23,24,25,26,27,28,29,30,31,32,33,34,35,36,37,38]. The crucial role of the lymphatics in keeping the central nervous system (CNS) healthy is supported by many facts. The development of vascular catastrophes, such as stroke, is accompanied by the activation of brain clearance from toxic blood products via the cervical lymphatics [29]. The blockade of this lymphatic pathway aggravates the severity of brain edema and contributes to an elevation of intracranial pressure after stroke [29,39]. The obstruction of nasal lymphatics due to inflammation could lead to a viral invasion and infection of the brain [35]. The surgical removal of deep cervical lymph nodes (dcLN)—the first anatomical connective bridge between the cerebral and peripheral lymphatics—leads to cognitive impairment in mice [40] and necrotic changes in neurons in rabbits [41]. Impairment of meningeal lymphatics induces cognitive impairment in mice suggesting that meningeal lymphatic dysfunction may be an aggravating factor in Alzheimer’s disease pathology and in age-associated cognitive decline [34].

However, studies of the role of the lymphatic system in brain clearance and recovery are still in its infancy. But, for “elusive” lymphatics, the new golden era is coming with novel conceptions about the anatomy and physiology of the lymphatic vascular system in the brain. Recent findings in this field call for a revision of our knowledge about the role of lymphatics in brain functions.

In this review, we are looking for an “Ariadne’s thread” analyzing contradictions between old and new hypothesis explaining the lymphatic anatomy and functions in the brain with a novel view on the interrelationships between the blood–brain barrier (BBB) and the cerebral lymphatic.

## 2. Where and How Lymph is Generated in the Brain?

The lymphatic system is part of the vascular system that carries a clear fluid called lymph (from Latin, lymph meaning “water”) [42]. During the day, approximately 20 L of plasma is filtered through the vascular endothelium of capillaries in the extracellular space in humans and 17 L of the filtered plasma are reabsorbed back into the venous vessels, while three liters remain in the extracellular or interstitial fluid (ISF) [43]. The lymphatic system is not a closed system and the lymphatic vessels are opened in the extracellular scape which provides the return of three liters of ISF to the blood.

Thus, in peripheral tissues, ISF is produced by filtration from plasma into extracellular space across the semipermeable vascular endothelium and then ISF is filtered back into the lymphatics, which returns it into the venous system. This means that the peripheral lymph is a fraction of ISF.

The situation with the brain is different due to the BBB restriction of fluid filtration from plasma into the brain. Also, the skull is incompressible, and the volume of brain fluid inside the skull is fixed.

However, in neurophysiology, there is no common agreement about the ISF formation in the brain. Abbott [15] proposed three models: (i) In the first model, ISF is produced by brain metabolism [44]; for example, the oxidation of glucose to carbon dioxide with the generation of water. But, the volume of metabolic ISF is too small (28 nlg^−1^ min^−1^ and around 10% of the total volume of ISF is 280 mL). (ii) In the second model, the generation of ISF is the cerebral capillary secretion of solutes, which are moved passively through the endothelial cell membranes or via the tight junctions into the connective system of perivascular space from the side of arteries/arterioles to veins/venules around neuropil and axon tracts (Figure 1) [15]. (iii) The third model explains ISF as a fraction of recycled CSF (cerebral spinal fluid), which flows from the choroid plexus into the subarachnoid space and then into PVS (perivascular space) where CSF are merged with ISF generated by cerebral capillaries [15].

It is interesting to note that sea animals (cuttlefish, mollusks, octopus, squids) who have no ventricles and CSF, have ISF with a similar flow rate of 0.1 µL g^−1^ min^−1^ as mammalians (rats, rabbits) [45]. This fact allows us to assume that capillary secretion of solutes might be a major part of ISF. However, this hypothesis needs more confirmations.

Taking together, these facts suggest that the most likely source of the lymph in the brain is two mixed components of brain fluids such as ISF and recycled CSF. However, it needs further detailed studies of the lymph formation from a fraction of brain fluids such as ISF and CSF.

## 3. The Gaps in the Lymphatic Anatomy in the Brain

We first learned about the lymphatics in the meninges through experiments of the Italian anatomist Paulo Mascagni [46] who described a network of transparent vessels in the layers of the brain already in the 18th century [31,46,47]. He also developed a new technique for the meningeal lymphatic visualization using the intracisternal injection of mercury. Today Mascagni’s models of the meningeal lymphatic vessels are presented in the Josephinum Wax Models Museum in Vienna [48]. However, Mascagni’s discovery of the meningeal lymphatics was forgotten for two centuries because no one could repeat his experiments due to technical problems. He was even regarded by several scientists as utopian: “Mascagni is so impressed with the lymphatics that he sees the lymph vessels everywhere even where they did not exist, i.e., in the brain” [47]. Therefore, it was the common belief about the lack of lymphatic circulation in the brain despite the growing body of qualitative and quantitative evidence supporting an integration between the brain fluids and lymphatics. Throughout the 19th century several groups demonstrated a functional connection between CBF and the lymphatics of the head. Nowadays, due to progress in optical techniques, Mascagni’s results are confirmed in the two independent works of Aspelung et al. [21] and Louveau et al. [22]. In these studies, it was clearly demonstrated that the meningeal lymphatics express all of the molecular hallmarks of lymphatic endothelium (Table 1). Afterwards, in other studies, the maturation of the meningeal lymphatics [49] and its role in brain drainage has been shown [32,33].

But the lymphatic anatomy still have more open questions than answers. Following Mascagni’s data, Aspelung et al. [21] and Louveau et al. [22] showed the lymphatic vessels lining the dural sinuses in the subarachnoid space but not in the brain tissues. In this aspect, it seems wrong to use the term “cerebral or brain lymphatics”, as used in many related publications [7,20,22,28,63] because lymphatic system in the brain parenchyma is not identified yet.

In 1979 Prineas [64] was the first who reported the appearance of the lymphatic capillaries and lymphoid tissues directly in the brain and spinal cord with neurological disorders. Thirty-seven years later, a group of German anatomists found the lymphatic vessels in deep areas of telencephalic hemispheres of healthy mice [1].

It is fundamentally important to fill the gaps in the anatomy of lymphatics in the brain to find Ariadne’s thread in the labyrinths of hypotheses about the physiology of lymphatics in the brain.

## 4. Problems with Current Concepts about Lymphatic Functions in the Brain

As we will discuss in the second section, the main function of the peripheral lymphatics is drainage of ISF, which is generated due to the filtration of fluids through the vascular endothelium. However, fluid filtration in the brain is restricted by BBB. Therefore, the classical knowledge about lymphatic drainage is not suitable for an explanation of functions of the cerebral lymphatic system that requests development of novel concepts explaining the elimination of fluids from the brain.

There are several theories explaining the drainage of brain fluids via different extracerebral lymphatic pathways [2,3,4,5,6,8,9,10,11,12,13,14,15,16,17,18,19,23,24,25,26,27,28], which are revised now in the light of re-discovery of the meningeal lymphatics [21,22]. Here we give a systematization and analysis of old and novel conceptions about brain drainage pathways with a modern view on the role of “cerebral” and peripheral lymphatics in this process.

The common view on brain fluids motion is based on the drainage of CSF as the main process. Cerebral spinal fluid is formed in the choroid plexus at a rate of 350 µL/min in humans (the total volume of 100–140 mL [65,66,67]) and 0.32 µL/min in mice (the total volume 35 µL mL [65,66,67]) and flows through the system of the four ventricles in the subarachnoid space (SAS), which is the final side of CSF reabsorption in the venous circulation (Figure 1). The main reabsorption of CSF occurs through the arachnoid villi and granulations that projects into the dural venous sinuses and additionally through the nasal lymphatics or along cranial nerve sheaths that play a supporting role.

But, there are some problems with this concept. The mechanisms underlying absorption of CSF from SAS into the bloodstream has remained speculative over many decades of investigations. Jonston et al. [11] discussed possible connective pathways between CSF and the arachnoid structures such as cell phagocytosis or pinocytosis via giant arachnoid vacuoles or passive transport through the extracellular arachnoid labyrinths (open tubes). However, whether arachnoid elements absorb CSF is unknown and some arachnoid projections are not associated with dural sinuses [11]. Furthermore, the arachnoid system does not appear to exist prenatally. At around the time of birth, the arachnoid villi and granulations start to become visibly, in infants, they increase in number and only in adults do they exist in abundance [68]. Therefore, the drainage of CSF via the arachnoid projects may be important in adulthood. Nevertheless, the choroid plexus produces CSF from the third gestational month, which suggests that the fetuses and neonates need an effective absorption of CSF [69]. It seems that other mechanisms excepting the arachnoid system might be important for drainage of CSF.

Weller et al. [16] discussed that there are significant differences between the drainage of CSF in animals and humans. So, up 50% of CSF in most animals drain into extracerebral lymphatics, while in humans it drains directly into venous circulation through the arachnoid system [16,19]. The proportion of CSF drainage into lymphatics remains unknown [70].

The concept about the role of the nasal lymphatics and the cribriform plate as a main structure in this brain drainage pathway was demonstrated in 1912 in humans [71] and was later confirmed in several animal studies that used an injection of different tracers into the CSF (the ventricles or the SAS) or the brain parenchyma and observation of tracers in the dcLN—the first anatomical station of CSF exit from the brain [10,17,18,19,23,24,25,26,72]. The observed time for the CBF drainage via the cervical lymphatic system in animals is presented in Table 2.

However, in a recent study of Lohrberg M. and Wilting J. [1] with using of a serial paraffin section of mouse head and specific labeling of lymphatic endothelium by Lyve-1 and Prox-1/CD31 showed missing of the lymphatic system in the nasal mucous membrane excepting the basal part of the interior turbinates [1]. They hypothesized that the nasal lymphatics is not the main pathway for drainage of CSF and mostly plays the role of moisturization of the respiratory air. They proposed a new pathway for the main lymph drainage pathway along the nasolacrimal duct (NLD) and the optic nerve connected with the eyelids and conjunctiva, which further drain lymph into the cervical lymph nodes. The NLD is presented in the embryonic period [73] and can be an alternative pathway for the arachnoid villi in embryos and neonates, which do not have the well-developed arachnoid system. These findings support the view of drainage of CSF via the arachnoid villi along the optic nerve of eyeballs. To confirm this theory, the authors discuss the presence of metastasis of ocular tumors in the cervical lymph nodes that might be explained by drainage of metastatic cells [74,75,76].

The ISF drainage pathways still remain an unclear and highly debated issue due to technical problems of quantitative measurements of ISF movement in very small extracellular spaces (ESC); i.e., the extracellular space in the in vivo rat cortex lies between 38–64 nm [77].

Recently the glymphatic system was proposed as a possible pathway for ISF and waster clearance of the brain [26,27,28]. Glymphatics utilizes the PVS tunnels, which are connected with astroglial cells to promote an elimination of metabolic proteins from the brain. The hypothesis about glymphatics is based on experimental data using labeling of CSF with fluorescent tracers injected into the cisterna magna. The tracers (dextran 2000 kDa and 3 kDa) were observed along the cortical pial arteries 100 μm below the cortical surface in 10–30 min after their injection with further diffusion into the brain parenchyma and elimination from the brain along the cerebral veins [26]. Using radiolabeled amyloid β1-40 that was injected into the brain parenchyma the authors showed that aquaporin-4 (AQP4) knockout mice revealed a significant reduction of the CSF flux and clearance of β-amyloid. It was, therefore, proposed that ISF drains by an AQP4-depended bulk flow through the brain parenchyma from the PVS of the cerebral arteries to the PVS of cerebral veins. Therefore, they proposed term “Glymphatics”, i.e., ISF drainage through glia cells.

However, Abbott [15] stated that the neuropil has a very narrow space between cells that makes it difficult to permit significant bulk flow, especially proteins. If radiolabeled traces (polyethylene glycols, and albumin) are cleared by diffusion, the small molecules would be expected to be cleared faster [78]. But, in fact, both small and large molecules are cleared in similar rates. So, Cserr and coworkers using tracers with a range of molecular weights showed that they are all cleared with a single rate constant [8,78,79,80]. They argued that these facts are against diffusion as a clearing mechanism. If diffusion is the dominant process, small and large molecules should have individual effective diffusion coefficients.

Cserr et al. [78,79] discussed that the PVS or the Virchow–Robin space surrounding the capillaries restricts the bulk flow. Abbott [15] in her review of bulk flow also provided that the Virchow–Robin space could be a significant mechanism for ISF drainage that, however, is hard to quantify. The osmotic gradient might be a driven pressure for solutes movement, where PVS is a low resistance pathway, while extracellular spaces in the neuropil are too narrow to permit a fluid flow. However, some authors state that diffusion through extracellular space (ECS) provides ISF drainage [30,81,82].

## 5. Mechanisms of ISF Fluid Transport in the Brain

One of the important factors that defines ISF transport speed in ECS is the relative proportion of the extracellular volume. Note that the study of the structure of ECS and the assessment of its volume are complicated by the fact that these characteristics may change significantly during the preparation of tissue samples. Thus, early measurements gave an ECS fraction of about 5% [83,84,85]. However, as was established later [86,87], this was the effect of the specific tissue preparation technology, and now it is assumed that ECS occupies from 10% to 30% of the total volume.

In the classical understanding of diffusion as a physical process, changes in ECS would affect the current concentration of ISF components, but not the speed of their propagation. However, the actual process in the parenchyma is much more complicated. It may involve the temporary binding of substances to receptors or elements of the extracellular matrix, as well as its temporary trapping [82].

A change in the extracellular volume may alter these processes. So, there are many reasons to expect a strong dependence of the transport rate on the current value of ECS. In this regard, recent data based on the dynamic regulation of the extracellular volume are of great importance.

It has long been known that the alternations of neural activity lead to a noticeable change in ionic gradients [88], and therefore, to the osmotic flow of water. But, neurons were found to be relatively resistant to osmolarity differences in terms of the regulation of their volume. Pasantes-Morales and Tuz [89] reported that hypo-osmolarity causes a change in the degree of excitability of brain cortex neurons, but has little effect on their volume, a sharp change of them is usually associated with cell death. Zhou et al. [90] described a rapid but reversible change in the neural volume with spreading depolarization, which can be classified as an extreme physiological state.

However, with astrocytes the situation is different. Risher et al. [91] showed that the volume of astrocytes quickly followed changes in the ionic composition of ISF, as expected, through a large number of functional aquaporins (AQPs) in their plasma membrane, which neurons do not have. Florence et al. [92] have found that in hippocampal slices the measurements reveal small changes of astrocyte image area, about 1% over 40 min, but a relatively small physiological increase in the concentration of extracellular potassium from 2.5 to 5 mM causes almost a 20% increase of the astrocyte volume. Pannasch et al. [93] noted that a regulation of the volume in the network of astrocytes is different from what a single cell shows, due to the redistribution of locally absorbed potassium over many coupled cells. Murphy et al. [94] discussed the contribution of the regulation of the astrocyte volume in the generation of epileptic seizures. They concluded that during stress the same basic regulatory mechanisms as glutamate uptake, extracellular potassium buffering, and brain water regulation that provide the tight junction control over neuronal excitability, may also be actively involved in seizure generation.

In the light of the above, it becomes evident that astrocytes, rather than neurons, are mainly responsible for modulating the volume of ECS in response to, for example, аdrenergic signaling, which triggers rapid changes in neural activity, which in turn can modulate the volume of ECS [95].

New data on the dynamic regulation of astrocyte volume were obtained in connection with attempts to clarify the differences in the intensity of drainage processes during sleep and wakefulness. It was found that astrocytes are responsible for the observed sensitivity of the extracellular volume to change of the ionic composition (rather than neuronal activity) that can be observed during sleep–wake transitions. Ding et al. [96] showed that ECS increased by more than 30% when artificially prepared “sleep” CSF was applied to the mice cortex, despite the mice remaining awake and mobile.

Evidently, all this should strongly affect the process of metabolite clearance from the brain. As reported in Xie et al. [97], “natural sleep or anesthesia are associated with a 60% increase in the interstitial space, resulting in a striking increase in the convective exchange of CSF fluid with ISF”.

In general, a detailed understanding of how quantitative, and possibly qualitative (topology, connectivity) characteristics of ECS vary depending on the mode of functioning of the brain, is extremely important in connection with the discussion of recently proposed alternative drainage mechanisms.

## 6. Conceptual Problems in Glymphatic Mechanisms of ISF Transport in the Brain

The proposed hypothesis about the glymphatic system and its role in the transport of ISF successfully linked a number of experimental observations and suggested a rather simple mechanism. For this reason, it was readily accepted and various publications appeared based on the glymphatic mechanism.

Diem et al. [98] suggested a computational model in order to describe the process of periarterial drainage in the context of diffusion in the brain. This model shows that the periarterial drainage along basement membranes is very rapid compared with diffusion.

Nakada et al. [99] discussed the specific organization of flows through AQPs. However, his hypothesis is only partially consistent with the originally proposed mechanism of glymphatics.

Bezerra et al. [100] proposed that the glymphatic dysfunction is identified as a major pathogenetic mechanism underpinning idiopathic intracranial hypertension.

However, a critical examination of the glymphatic hypothesis indicates that not all links of the mechanism are explained unambiguously. In particular, the proposed propulsive function of pulsations of penetrating arterial vessels was questioned. In the comprehensive review by Hladky and Barrand [101], several possible options for the organization of flows are considered, being alternatives to the glymphatics.

While the presence of pulsations can be regarded as a proven fact [102,103], their ability to create a directional fluid flow along the arterial PVS is not obvious. In Asgari et al. [104], an analysis of the proposed glymphatic mechanism by mathematical modeling was carried out and it was concluded that the presence of a bulk flow is doubtful and that dispersion, rather than convection, is the most probable mechanism for transporting tracer to the parenchyma. Diem et al. [105] came to the conclusion that arterial pulsations cannot drive intramural periarterial drainage. Smith et al. [106] detected the declared flow patterns in the parenchyma, but they also found that (i) the transport of fluorescent dextrans in brain parenchyma depends on the dextran size in a manner consistent with diffusive rather than convective transport; (ii) transport of dextrans in the parenchymal ESC, measured by 2-photon fluorescence recovery after photobleaching, was not affected just after cardiorespiratory arrest; and (iii) AQP4 gene deletion did not impair a transport of fluorescent solutes from SAS to the brain of mice or rats. In a further work, Smith et al. [107] concluded that “the theoretical plausibility of glymphatic transport has been questioned, and recent data have challenged its experimental underpinnings”.

One of the serious reasons for doubts in the propulsive work of the pulse wave is related with its wavelength, which can be estimated on the basis of the previously reported pulse wave velocity for small vessels, which is as low as 10 cm per second [108,109]. With this, the length of the pulse wave is more than ten times larger than the working distance, which is less than 1 mm. Therefore, the cardiac rhythm should cause an almost simultaneous, non-directional change in the volume of PVS, rather than a running pulse.

In this regard, a discussion of the role of AQP4 also looks ambiguous. There is a general agreement on their important role in the flow of water in the parenchyma, including the dynamic regulation of the astrocyte volume, as mentioned above. In the framework of the glymphatic hypothesis, they are assigned the role of the main conductor of the glymphatic flow. Indeed, in the work of Asgari et al. [110], the fundamental possibility of such a flow through the astrocyte network was shown, but it was presumed that there is a pressure gradient. Nakada et al. [99] hypothesized that AQPs deliver water to the near-capillary region but the fluid flow there is significantly limited by BBB. This hypothesis needs to be further justified since the trans-network transmission of a considerable amount of water might overload the mesh of thin astrocyte processes, where the gap junctions between astrocytes are located.

Bacyinski et al. [111] concluded that there is currently significant controversy in the literature regarding both the direction of waste clearance as well as the pathways by which the waste-fluid mixture is cleared. Benveniste et al. [112], the inconsistencies in the data of various papers are analyzed, pointing to the imperfection of experimental techniques.

Up to date, there is a number of recently published studies that support the idea that perivascular transport may include the directed flows, but the exchange between CSF and ISF is likely diffusive [105,106,107,113,114]. This viewpoint is summarized in the recent review paper by Abbott et al. [115], where they argue that recent evidence suggests important amendments to the “glymphatic” hypothesis.

It worth noting that since the glymphatic mechanism was found as a suitable paradigm, it is still used “as is” in a number of studies on specific problems related with brain drainage, such as impairment of drainage during diabetes [116], inverse correlation with state-dependent lactate concentration in the brain [117], its impairment during multiple microinfarcts [118], or in a mouse model of Alzheimer’s disease. Such recent applications of the glymphatic hypothesis are recently reviewed by Plog and Nedergaard in Reference [119].

In summary, the existence of a perivascular fluid system, whereby CSF enters the brain via flow through PVS is supported both by new as well as key historical studies. The specific moving pattern of fluid in PVS, directed or oscillating with a low or vanishing net flow, still needs to be justified, but the latter one seems to be more consistent with physical laws. This is what is referred to as “dispersion” being the combination of a directed flow together with diffusion. This mechanism, even with zero net flow, is able to provide a much faster transport of different substances, that can be achieved with the conventional diffusion process. Interestingly, this mechanism that recently gained much attention, is consistent with the 27-year-old observation of Ichimura et al. [80] who observed that the direction of the flow was variable, with a vector into the brain along one segment of an artery and out of the brain in a more distal segment.

In the light of BBB-related issues, every aspect discussed above is important, since it can appear, that substances, which penetrate the opened BBB will be transported to the surface layers of the cortex and further to the lymphatic vessels, rather than to the deep parts of the parenchyma.

Potentially, this is the second challenge after overcoming BBB in solving the problems of drug delivery, and therefore, progress in understanding this issue is in great demand.

## 7. Alternative Notions for Lymphatic Functions in the Brain

Based on theoretical analysis of hypothesis about lymphatic functions in the brain, Czerr [18] was the first, who proposed in 1992 an intriguing idea about the relationship between BBB and lymphatic drainage of the brain. However, only recently, has his idea been supported experimentally [32,33,120].

Here we discuss our original data demonstrating a connective bridge between the BBB opening and activation of the meningeal lymphatic clearance. We used two different methods for BBB opening. The first method, photodynamic (PD) opening of BBB, induces a strong BBB disruption associated with vasogenic edema and a significant increase in the CSF volume [32,121]. The second method, a loud audible sound, causes a mild BBB opening that is not accompanied by visible changes in PVS [33,122] or in the CSF volume [120]. Both cases of BBB disruption were accompanied by an increase in the diameter of the meningeal lymphatic vessels that was much more pronounced in the PD group vs. the sound group. These results suggest that lymphatic changes were equivalent to the severity of BBB disruption and to the volume of CSF [122]. 

Using sound and PD to open BBB for FITC-dextran 70 kDa in experiments ex vivo and in vivo on healthy mice, we revealed that the meningeal lymphatics is the pathway in the brain clearing after the BBB opening [32,33]. Indeed, 20 min after the FITC-dextran extravasation from the cerebral vessels into the brain tissues, a tracer was observed in the meningeal lymphatic vessels. We also demonstrate the functional connection between the meningeal and cervical lymphatics, which are both involved in brain clearing from molecules that cross the opened BBB. Using optical coherent tomography and gold nanorods (GNRs) as contrast agents, we monitored the accumulation of GNRs in the dcLN after their crossing of opened BBB by sound (mild BBB opening) and PD (strong BBB opening). These data clearly demonstrate, the stronger BBB was disrupted, the faster GNRs accumulated in the dcLN.

Aspelung et al. [21] also showed the involvement of the meningeal lymphatics in the brain clearing from high molecular weight molecules. This group was not focused on the study of interactions between BBB and the meningeal lymphatics but they used an injection of tracers directly into the brain parenchyma that might destroy BBB structure and stimulate brain drainage as well.

Thus, our results give initial hints for a connective bridge between the BBB opening and activation of meningeal lymphatic clearance (Figure 2). These novel findings may call for a reassessment of the basic assumption in the mechanisms underlying the brain recovery after events associated with the BBB disruption such as hemorrhagic stroke, subdural and subarachnoid hemorrhages, brain trauma, edema, and neurodegenerative diseases.

To confirm our idea, we analyzed the clearance of the brain from toxic products of blood in patients who have died after different hemorrhagic events using an immunohistochemical assay and atomic absorption spectroscopy (non-published data). In all cases, we found the presence of the hemosiderin/iron in the meningeal lymphatic system and in the dcLN.

These data shed light on the role of the meningeal lymphatics in the mechanisms underlying the neuropathology and open novel strategies for a non-invasive stimulation of lymphatic brain drainage by use of lasers (see Review “Non-invasive photonic technologies with computer adaptive control for neurorehabilitation therapy”).

In summary, a better understanding of anatomy and physiology of the lymphatics in the brain will give new knowledge about the role of lymphatics in the rehabilitation of the brain functions after neural pathologies. The ability to stimulate lymph flow in the brain is likely to play an important role in developing future innovative strategies in neurorehabilitation therapy.

## Figures and Tables

**Figure 1 ijms-19-03818-f001:**
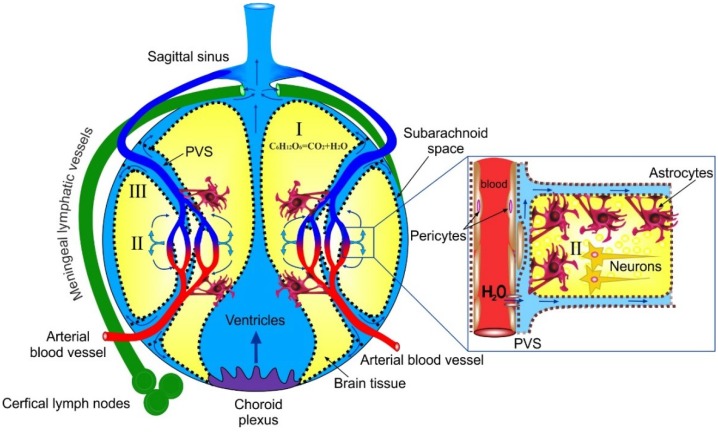
Schematic illustration of three models of generation and pathways of interstitial fluid (ISF) in the brain. Model I explains the formation of ISF (10% of the total volume of ISF) as a result of metabolic oxidation of glucose to carbon dioxide and water; Model II explains the generation of ISF as a large fraction of cerebral capillary secretion of solutes, which are driven passively by the ionic gradient through the endothelial cell membranes (blue arrows) or via the tight junctions of (blood-brain barrier (BBB) and formed perivascular space (PVS—enlarged figure; arrows show movement of ISF) around penetrating arteries, venules, and veins, and connecting with glia-lines boundaries between neuropil and regions of axon tracts; Model III explains ISF as a fraction of recycled cerebral spinal fluid (CSF), which flows from the choroid plexus into the subarachnoid space and then into PVS where CSF is merged with ISF generated by cerebral capillaries.

**Figure 2 ijms-19-03818-f002:**
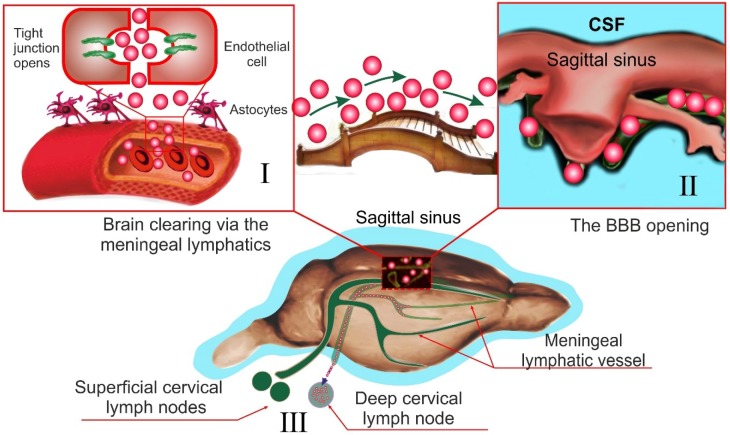
A schematic illustration of connective bridge between the BBB opening (on the example of FITC-dextran) and activation of the lymphatic clearance: I illustrates the BBB opening for FITC-dextran (the red bolls); the green arrows show movement of the extravasated FITC-dextran from the brain tissues into the meningeal lymphatic vessels; II demonstrates movement of FITC-dextran in the cerebral lymphatic network; III show that after the BBB opening, FITC-dextran moves from the meningeal lymphatics (II) into the deep cervical lymph nodes (the red dotted boxes).

**Table 1 ijms-19-03818-t001:** The molecular hallmarks of the lymphatic endothelium.

The Markers of the Lymphatic Endothelium	The Functional Characteristics of Proteins Expressed in the Lymphatic Endothelium
LYVE-1—Lymphatic vessel endothelial hyaluronan receptor 1	Hyaluronan (HA) is an element of skin and mesenchymal tissues that regulates cell migration in the course of wound healing, inflammation, and embryonic morphogenesis [50]. LYVE-1 is a specific transmembrane receptor of HA, which was first described by Banerji in 1999 [51]. LYVE-1 was found primarily on both the luminal and abluminal surface of lymphatic endothelial cells [51,52]. The functional role of LYVE-1 is still the subject of debate, but there are evidences that LYVE-1 plays an important role in hyaluronan transport and turnover, or in hyaluronan localization to the surface of lymphatic endothelium providing for migration CD44+ leukocytes or tumor cells [53]. Notice, there are some findings where expression of Lyve-1 was also observed in the human iliac atherosclerotic arteries [54], in the embryonic blood vessels [55], in macrophages [56], the reticulo-endothelial system [48]. Therefore, the specificity of Lyve-1 as a marker for the lymphatic vessels is not strong enough.
Prox1—Prospero homeobox protein 1	Transcription factor regulating the process of growth and differentiation of endothelial cells of lymphatic vessels [57]
CCL21-Chemokine (C-C motif ligand 21)	It is secreted by endothelial cells of lymphatic vessels and is involved in activation of T-lymphocyte movement, migration of lymphocytes to other organs, and dendritic cells into lymph nodes [58].
VEGFR3—Vascular endothelial growth factor receptor 3	VEGFR3 is a receptor that triggers the lymphangiogenesis, i.e., the formation of new lymphatic vessels [59]. In transgenic mice with absence of *VEGFR3* gene, the meningeal lymphatic vessels do not develop, and lymph node hypoplasia is noted [21].
PDPN—Podoplanin	PDPN is integral membrane protein, which is responsible for the normal development of the network of lymphatic vessels, providing drainage of the intercellular fluid. If the synthesis is broken, lymphedema is formed [60]. The PDPN activation is accompanied by lymphangiogenesis, which is regarded as an important indicator of tumor growth [61].
ITGA9—Integrin-α9	ITGA9 is a protein, which is a part of the valves in the lymphatic vessels [62].

**Table 2 ijms-19-03818-t002:** The time for the CBF drainage via the cervical lymphatic system in animals [18,72].

Objects	The Side of Injection of Tracer (Radio-Iodinated Albumin)	Time of Lymph Collection (h)	Lymph Recovery (%) *
Rabbit	Caudate nucleus	25	47
	Internal capsule	25	22
	Brain	25	18
	CSF	6	30
Cat	CSF	8	14
Sheep	CSF	26	32

* The lymph recovery is given as percent of the total lymph outflow from the central nervous system.

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
