# Peer review of "Blood–Brain Barrier, Lymphatic Clearance, and Recovery: Ariadne’s Thread in Labyrinths of Hypotheses"

_ijms, 2018, doi:10.3390/ijms19123818_

Round 1
Reviewer 1 Report
The recent advances in imaging have revealed very intriguing observations of the brain lymphatics, much of it completely change our traditionally thinking of this "immune-privileged" site. Although I feel this review is timely, it was poorly written. The authors came up with the tongue-in-cheek title "Ariadne's thread in labyrinths of hypothesis", which this reviewer appreciates, but the review was written in a confused and disorganised manner, leaving me more confused than ever. The review would have improved immensely with proper planning, sub-sectional titles and tables with either lines or different colours between rows. Furthermore, the manuscript was submitted to a general science journal but the review was not structure for readers outside the field. Importantly, it is not acceptable to review another review article - please present the evidence from the original research papers and form a coherent argument.
Author Response
The detailed response to Referee 1
Comment: Although I feel this review is timely, it was poorly written. The authors came up with the tongue-in-cheek title "Ariadne's thread in labyrinths of hypothesis", which this reviewer appreciates, but the review was written in a confused and disorganised manner, leaving me more confused than ever. The review would have improved immensely with proper planning, sub-sectional titles and tables with either lines or different colours between rows. Furthermore, the manuscript was submitted to a general science journal but the review was not structure for readers outside the field.
Response: We would like to thank the Referee for the very useful comments and recommendations. In the revised version of the manuscript, we implemented all your suggestions made as specified below:
“..review would have improved immensely with proper planning, sub-sectional titles..”
We re-arranged the whole text; it is now divided into 7 sections, which are, in our opinion, more logically related with each other. Namely:
In the first section entitled “Lymphatics and central nervous system health” we start with a discussion about the major role of the cerebral and cervical lymphatics in keeping the central nervous system (CNS) in a healthy state.
In the second section “Where and how lymph is generated in the brain?” we address different hypotheses on lymph formation in the periphery and in the brain. Figure 1 illustrates the brain-related hypothesis about the generation of the interstitial fluid as a main component of lymph.
In the third section “The gaps in the lymphatic anatomy in the brain” we present some historical facts and modern concepts about the lymphatic anatomy with the focus on novel findings, which suggest a substantial revision of our knowledge about the anatomy of the lymphatic system in the brain. In table 1 we list the proteins of the lymphatic endothelium and discuss their functions.
In the fourth section “Problems with current concepts about lymphatic functions in the brain” we consider the lymphatic functions in the brain showing the limitations of old hypotheses about drainage and clearance of the brain fluids.
In fifth and sixth sections “Mechanisms of ISF fluid transport in the brain” and “Conceptual problems in glymphatic mechanisms of ISF transport in the brain” we analyze recently revealed contradictions in the currently popular hypothesis of fluid motion mechanisms in the brain.
In seventh section “Alternative notions for lymphatic functions in the brain” we discuss the novel in vivo and ex vivo data, which that allow one to “make a bridge” between the opening of blood-brain barrier and the subsequent activation of the meningeal lymphatic clearance that might be the major mechanism underlying the brain recovery after neurological pathologies such as stroke, brain trauma, hemorrhagic events, brain edema, and neurodegenerative diseases. During the revision, we have changed Figure 2 in order to present our idea about the relationship between the opening of the blood-brain barrier and the meningeal lymphatic clearance in a more clear way.
In summary, we draw the conclusion that a better understanding of anatomy and physiology of the lymphatics in the brain will provide new insights about its role in the rehabilitation of the brain functions after neural pathologies. The ability to stimulate the lymph flow in the brain, is likely to play an important role in developing future innovative strategies in neurorehabilitation therapy.
Thus, all sessions now provide a logical transition from anatomy to functions of the lymphatic system in the brain with a discussion of a limitations of old hypothesis and contradiction of modern concepts in this field. The style of review allows readers to be included in dialogue between old and modern era for elusive lymphatics in the brain exciting them to wander in labyrinths of different ideas with Ariadne’s thread to find novel strategies for promising therapeutic tools in neurorehabilitation.
Comment: “... tables with either lines or different colors between rows.”
Response: We have revised the Tables (Lines 102-104 and 159-161) as shown below:
Table 1. The molecular hallmarks of the lymphatic endothelium.
LYVE-1 - Lymphatic vessel endothelial hyaluronan receptor 1 | Hyaluronan (HA) is an element of skin and mesenchymal tissues that regulates cell migration in the course of wound healing, inflammation and embryonic morphogenesis [49]. LYVE-1 is a specific transmembrane receptor of HA, which was first described by Banerji in 1999 [50]. LYVE-1 was found primary on both the luminal and abluminal surface of lymphatic endothelial cells [50, 51]. The functional role of LYVE-1 is still the subject of debate, but there are evidences that LYVE-1 plays an important role in hyaluronan transport and turnover, or in hyaluronan localization to the surface of lymphatic endothelium providing for migration CD44+ leukocytes or tumor cells [52]. Notice, there are some findings where expression of Lyve-1 was also observed in the human iliac atherosclerotic arteries [53], in the embryonic blood vessels [54], in macrophages [55], the reticulo-endothelial system [56]. Therefore, the specificity of Lyve-1 as a marker for the lymphatic vessels is not strong enough. |
Prox1 - Prospero homeobox protein 1 | Transcription factor regulating the process of growth and differentiation of endothelial cells of lymphatic vessels [57] |
CCL21-Chemokine (C-C motif ligand 21) | It is secreted by endothelial cells of lymphatic vessels and is involved in activation of T-lymphocyte movement, migration of lymphocytes to other organs and dendritic cells into lymph nodes [58]. |
VEGFR3 - Vascular endothelial growth factor receptor 3 | VEGFR3 is a receptor that triggers the lymphangiogenesis, i.e. the formation of new lymphatic vessels [59]. In transgenic mice with absence of VEGFR3 gene, the meningeal lymphatic vessels do not develop and lymph node hypoplasia is noted [21]. |
PDPN - Podoplanin | PDPN is integral membrane protein, which is responsible for the normal development of the network of lymphatic vessels, providing drainage of the intercellular fluid. If the synthesis is broken, lymphedema is formed [60]. The PDPN activation is accompanied by lymphangiogenesis, which is regarded as an important indicator of tumor growth [61]. |
ITGA9 - Integrin-α9 | ITGA9 is a protein, which is a part of the valves in the lymphatic vessels [62]. |
Table 2. The time for the CBF drainage via the cervical lymphatic system in animals [18, 72].
Objects | The side of injection of tracer (radio-iodinated albumin) | Time of lymph collection (h) | Lymph recovery (%)* |
Rabbit | Caudate nucleus | 25 | 47 |
Internal capsule | 25 | 22 | |
Brain | 25 | 18 | |
CSF | 6 | 30 | |
Cat | CSF | 8 | 14 |
Sheep | CSF | 26 | 32 |
*- The lymph recovery is given as percent of the total lymph outflow from the CNS.
Comment: Importantly, it is not acceptable to review another review article - please present the evidence from the original research papers and form a coherent argument.
Response: We generally agree with this. We have expanded the discussion, where it was appropriate (say, about conflicting points of view, lines 382-393).
However, since our review includes 125 references, and among them, about 20% are review articles, it seems to be impossible to reproduce in full their argumentation. In fact, we cite these review papers as the original sources of motivated opinion, rather than as sources of experimental data revised in there. Let us note, the resulted “navigation in space of hypothesis” is consistent with a title of our review “Ariadne’s thread in labyrinths of hypothesis”.
Comment: “.. but the review was not structured for readers outside the field. “
Response: We hope, the improved manuscript structure will be helpful for a broad readership. In addition, we introduce the separate table with an explanation of abbreviations used through the text.
We would like to thank the Reviewer again for the constructive comments and suggestions that helped us a lot to substantially improve our manuscript.
Sincerely yours,
Authors

Reviewer 2 Report
This is a very timely review article as this is a growing field with exciting new data emerging. This manuscript is fairly comprehensive and includes historical references. These authors have published several papers recently on photodynamic BBB opening and downstream effects on meningeal lymphatic vessels and have the expertise to write a review article of this nature.
Comments:
Please define "recruitment of meningeal lymphatics".
Section 1
Lines 35-36-there is no reference provided for the statement saying that neurons die after about 5 minutes of oxygen deprivation. Other reviews state that this happens after 1 minute thus this statement needs to be changed or a reference needs to be provided.
Section 2-lines 80-93, there are no citations in these two paragraphs yet there are data provided. References are critical to add here.
Line 97-it is unclear if this conclusion was drawn in the Abbott review or if this is an opinion from the authors.
Sections 4 and 5 are too long and the flow of topics covered is not well connected. I recommend introducing subtopics to help the reader understand the points that are being made. Also, there are many 1-2 sentence paragraphs. These should be grouped into other paragraphs where it makes sense.
168-171 is a very good point.
180-need a reference number here.
207-213-This paragraph needs more explanation. Where were tracers injected? The point about diffusion and small molecules being cleared faster needs a reference or more description.
230-remove "and references there".
231-232-this should not be a separate paragraph.
233-234-this should not be a separate paragraph.
235- it is not clear to me what "intense neural activity" is referring to.
245-246- is this statement referring to data in the previous reference? The way it is worded, it doesn't read as though this is from the same body of work.
247-249-this should not be a separate paragraph.
323-327: expand on the information presented in these two conflicting reviews (Needergaard and Abbott).
363-"Alexander Monro and Kellie" is not this reference (31)
402-403-"high weight molecules" shld be "high molecular weight molecules".
425-426- what is atomic abortion spectroscopy?
Figure 2 figure legend needs to be more clear.
New data from Da Mesquita et al 2018 Nature should be mentioned somewhere is this article as it is relevant to the topic (abeta was brought up in the glymphatics section). "Functional aspects of meningeal lymphatics in ageing and Alzheimer's disease".
Author Response
The detailed response to Reviewer 2.
First, we would like to thank the Reviewer for the very constructive comments and helpful advice.
Comment: Please define "recruitment of meningeal lymphatics".
Response: During the revision of the manuscript, we have replaced the "recruitment of meningeal lymphatics" with the more detailed and self-explanatory phrasing “connective bridge between the BBB opening and activation of the meningeal lymphatic clearance”.
We also substantially revised section 5 (now it is section 7, lines 334-380 of the revised manuscript) and replaced Figure 2 in order to show more clearly the interrelationship between the BBB opening and activation of the meningeal lymphatic clearance. All changes in the manuscript are highlighted by yellow.
Comment: Section 1. Lines 31-36-there is no reference provided for the statement saying that neurons die after about 5 minutes of oxygen deprivation. Other reviews state that this happens after 1 minute thus this statement needs to be changed or a reference needs to be provided.
Response: In the revision of the manuscript, we have removed this statement. However, we would like to note, that we took this information from the comprehensive book “Human Physiology” (published in Russian). Also, a similar information can be found in the e-book “Function of cells and human body”:
http://fblt.cz/en/skripta/regulacni-mechanismy-2-nervova-regulace/3-energeticky-metabolismus-nervove-tkane/
”Shortage of oxygen causes unconsciousness within few tenths of seconds and the damage to neurons becomes irreversible after about 5 minutes”.
Comment: Section 2-lines 80-93, there are no citations in these two paragraphs yet there are data provided. References are critical to add here.
Response: Thank you for this recommendation. We added the following citations at lines 53 and 56:
"Lymph - Definition and More from the Free Merriam-Webster Dictionary". www.merriam-webster.com. Retrieved 2010-05-29.
Human Physiology From Cells to Systems 8th (eighth) Edition by Sherwood Lauralee: Nelson Education (2015).
“The lymphatic system is part of the vascular system that carries a clear fluid called lymph (from Latin, lymph meaning "water") [42]. During day approximately 20 liters of the plasma is filtered through vascular endothelium of capillaries in the extracellular space in human and 17 liters of the filtered plasma are reabsorbed back into the venous vessels, while 3 liters remain in the extracellular or interstitial fluid (ISF) [43]. The lymphatic system is not a closed system and the lymphatic vessels are opened in the extracellular scape that provide returning of 3 liters of ISF to the blood.”
Comment: Line 97-it is unclear if this conclusion was drawn in the Abbott review or if this is an opinion from the authors.
Response: We have revised the text in order to explain more clearly Abbott’s models of ISF formation in the brain (Line 65):
“However, in neurophysiology, there is no common agreement about the ISF formation in the brain. Abbott proposed three models [15]: i) the first model, ISF is produced by brain metabolism [44]. For example, the oxidation of glucose to carbon dioxide with the generation of water. But, the volume of metabolic ISF is too small (28 nlg-1 min-1around 10% of the total volume of ISF that is 280 ml); ii) the second model, the generation of ISF is the cerebral capillary secretion of solutes, which are moved passively through the endothelial cell membranes or via the tight junctions into the connective system of perivascular space from the side of arteries/arterioles to veins/venules around neuropil and axon tracts (Figure 1) [15]; iii) the third model explains ISF as a fraction of recycled CSF, which flows from the choroid plexus into the subarachnoid space and then into PVS where CSF are merged with ISF generated by cerebral capillaries [15]”.
Comment: Sections 4 and 5 are too long and the flow of topics covered is not well connected. I recommend introducing subtopics to help the reader understand the points that are being made. Also, there are many 1-2 sentence paragraphs. These should be grouped into other paragraphs where it makes sense.
Response: This has been corrected. This part of the manuscript now consists of 4 sections:
4. Problems with current concepts about lymphatic functions in the brain (lines 115-201)
5. Mechanisms of ISF fluid transport in the brain (lines 203-254)
6. Conceptual problems in glymphatic mechanisms of ISF transport in the brain (lines 256-332)
7. Alternative notions for lymphatic functions in the brain (lines 334-375)
And we have merged too short paragraphs.
Comment: 180-need a reference number here.
Response: There was the reference [13], in the current manuscript version it becomes reference [1] (Line 159).
“However, in a recent study of Lohrberg M. and Wilting J. with using of a serial paraffin section of mouse head and specific labeling of lymphatic endothelium by Lyve-1 and Prox-1/CD31 showed missing of the lymphatic system in the nasal mucous membrane excepting the basal part of the interior turbinates [1].”.
Comment: 207-213-This paragraph needs more explanation. Where were tracers injected? The point about diffusion and small molecules being cleared faster needs a reference or more description
Response: We added the description of the type of tracers (Lines 188-195):
“However, Abbott stated that the neuropil has a very narrow space between cells that make difficult to permit significant bulk flow, especially proteins [15]. If radiolabeled traces (polyethylene glycols and albumin) are cleared by diffusion, the small molecules would be expected to be cleared faster [78]. But, in fact, both small and large molecules are cleared in similar rates. So, Cserr and coworkers using tracers with a range of molecular weights showed that they all are cleared with a single rate constant [8, 78, 79]. They argued that these facts are against diffusion as a clearing mechanism. If diffusion is the dominant process, small and large molecules should have individual effective diffusion coefficients.”
[15] Abbott, N.J. Evidence for bulk flow of brain interstitial fluid: significance for physiology and pathology. Neurochem Int. 2004, 45, P. 545–552. doi: 10.1016/j.neuint.2003.11.006
[78] Cserr, H.F., Cooper, D.N., Suri, P.K., Patlak, C.S., 1981. Efflux of radiolabeled polyethylene glycols and albumin from rat brain. Am. J. Physiol. 240, 319–328
Comment: 230-remove "and references there".
Response: This has been corrected (Line 214).
Comment: 231-232-this should not be a separate paragraph. 233-234-this should not be a separate paragraph. 235- it is not clear to me what "intense neural activity" is referring to.
Response: The separate paragraphs have been merged. The phrase has been revised, "intense neural activity" and has been changed to “alternations of neural activity” and new reference has been added (lines 215-247).
“A change in the extracellular volume may alter these processes. So, there are many reasons to expect a strong dependence of the transport rate on the current value of ECS. In this regard, recent data based on the dynamic regulation of the extracellular volume are of great importance.
It has long been known that the alternations of neural activity lead to a noticeable change in ionic gradients [89], and therefore to the osmotic flow of water. But, neurons were found to be relatively resistant to osmolarity differences in terms of the regulation of their volume. Pasantes-Morales and Tuz [90] reported that hypo-osmolarity causes a change in the degree of excitability of brain cortex neurons, but has little effect on their volume, a sharp change of them is usually associated with cell death. Zhou et al. [91] described a rapid but reversible change in the neural volume with spreading depolarization, which can be classified as an extreme physiological state.
However, with astrocytes the situation is different. Risher et al. [92] showed that the volume of astrocytes quickly followed changes in the ionic composition of ISF, as expected, through a large number of functional AQPs in their plasma membrane, which neurons do not have. Florence et al. [93] have found that in hippocampal slices the measurements reveal small changes of astrocyte image area, about 1% over 40 minutes, but a relatively small physiological increase in the concentration of extracellular potassium from 2.5 to 5 mM, causes almost 20% increase of the astrocyte volume. Pannasch et al [94] noted that a regulation of the volume in the network of astrocytes is different from what a single cell shows, due to the redistribution of locally absorbed potassium over many coupled cells. Murphy et al. [95] discussed the contribution of the regulation of the astrocyte volume in the generation of epileptic seizures. They concluded, that during stress the same basic regulatory mechanisms as glutamate uptake, extracellular potassium buffering, and brain water regulation that provide the tight junction control over neuronal excitability, may also be actively involved in seizure generation.
In the light of the above, it becomes evident that astrocytes, rather than neurons, are mainly responsible for modulating the volume of ECS in response to, for example, аdrenergic signaling, which triggers rapid changes in neural activity, which in turn can modulate the volume of ECS [96].
New data on the dynamic regulation of astrocyte volume were obtained in connection with attempts to clarify the differences in the intensity of drainage processes during sleep and wakefulness. It was found that astrocytes are responsible for the observed sensitivity of the extracellular volume to change of the ionic composition (rather than neuronal activity) that can be observed during sleep-wake transitions. Ding et al. (2016) [97] showed that ECS increased by more than 30% when artificially prepared “sleep” CSF was applied to the mice cortex, despite the mice remaining awake and mobile.”
Comment: 245-246- is this statement referring to data in the previous reference? The way it is worded, it doesn't read as though this is from the same body of work.
Response: This and the following phrases have been revised (lines 225-227).
Comment: 247-249-this should not be a separate paragraph.
Response: This has been corrected (Lines 215-247).
Comment: 323-327: expand on the information presented in these two conflicting reviews (Needergaard and Abbott).
Response: The text has been revised. Now two conflicting reviews are presented in a different and more detailed manner (Lines 307-317).
“Up to date, there is number of recently published studies that support the idea that perivascular transport may include the directed flows, but the exchange between CSF and ISF is likely diffusive [106-108], [114, 115]. This viewpoint is summarized in the recent review paper by Abbott et al. [116], where they argue, that recent evidence suggests important amendments to the 'glymphatic' hypothesis.
It worth to be noted that since the glymphatic mechanism was found as a suitable paradigm, it is still used “as is” in the number of studies on specific problems related with brain drainage, such as impairment of drainage during diabetes [117], inverse corelation with state-dependent lactate concentration in the brain [118], its impairment during multiple microinfarcts [119], or in a mouse model of Alzheimer’s disease. Such recent applications of the glymphatic hypothesis are recently reviewed by Plog and Nedergaard in [120]”.
Comment: 363-"Alexander Monro and Kellie" is not this reference (31).
Response: We meant this reference: B. Mokri, “The Monro–Kellie hypothesis: applications in CSF volume depletion,” Neurology 56(12), 1746–1748 (2001). In the revised manuscript we remove this reference.
Comment: 402-403-"high weight molecules" should be "high molecular weight molecules".
Response: Thank you for your remark, we have made the correction (Line 358-359).
“Aspelung et al. also showed the involvement of the meningeal lymphatics in the brain clearing from high molecular weight molecules [21]”.
Comment: 425-426- what is atomic abortion spectroscopy?
Response: It was misprinted and has been corrected. It should be atomic absorption spectroscopy (Lines 367-370).
“To confirm our idea, we analyzed the clearance of the brain from toxic products of blood in patients died after different hemorrhagic events using of the immunohistochemical assay and atomic absorption spectroscopy (not published data). In all cases, we found the presence of the hemosiderin/iron in the meningeal lymphatic system and in the dcLN”.
Comment: Figure 2 figure legend needs to be more clear.
Response: Figure 2 and legend have been changed.
Comment: New data from Da Mesquita et al 2018 Nature should be mentioned somewhere is this article as it is relevant to the topic (abeta was brought up in the glymphatics section). "Functional aspects of meningeal lymphatics in ageing and Alzheimer's disease".
Response: Thank you so much for this important recommendation, we cited this interesting new publication (reference 34) in Sections 1 (Line 33).
Let us express again our appreciation that you found time to work with our review and for your kind assistance for the improvement of our article. It was a very good experience for us in preparation of this review with your help for “International Journal of Molecular Science”.
Sincerely yours,
Authors

Round 2
Reviewer 1 Report
No further comments.